# Somatostatin and Its Receptor System in Colorectal Cancer

**DOI:** 10.3390/biomedicines9111743

**Published:** 2021-11-22

**Authors:** Aldona Kasprzak

**Affiliations:** Department of Histology and Embryology, Poznan University of Medical Sciences, Święcicki Street 6, 60-781 Poznań, Poland; akasprza@ump.edu.pl

**Keywords:** colorectal cancer, neuroendocrine tumors, SRIF system, clinical application

## Abstract

Somatostatin (SST)/somatotropin release-inhibiting factor (SRIF) is a well-known neuropeptide, widely distributed in the central and peripheral nervous systems, that regulates the endocrine system and affects neurotransmission via interaction with five SST receptors (SST1-5). In the gastrointestinal tract, the main SST-producing cells include intestinal enteroendocrine cells (EECs) restricted to the mucosa, and neurons of the submucosal and myenteric plexuses. The action of the SRIF system is based on the inhibition of endocrine and exocrine secretion, as well as the proliferative responses of target cells. The SST1–5 share common signaling pathways, and are not only widely expressed on normal tissues, but also frequently overexpressed by several tumors, particularly neuroendocrine neoplasms (NENs). Furthermore, the SRIF system represents the only peptide/G protein-coupled receptor (GPCR) system with multiple approved clinical applications for the diagnosis and treatment of several NENs. The role of the SRIF system in the histogenesis of colorectal cancer (CRC) subtypes (e.g., adenocarcinoma and signet ring-cell carcinoma), as well as diagnosis and prognosis of mixed adenoneuroendocrine carcinoma (MANEC) and pure adenocarcinoma, is poorly understood. Moreover, the impact of the SRIF system signaling on CRC cell proliferation and its potential role in the progression of this cancer remains unknown. Therefore, this review summarizes the recent collective knowledge and understanding of the clinical significance of the SRIF system signaling in CRC, aiming to evaluate the potential role of its components in CRC histogenesis, diagnosis, and potential therapy.

## 1. Introduction

Colorectal cancer (CRC) is one of the most common human malignancies worldwide, with more than 1.9 million new cases and 935,000 deaths in 2020. Overall, CRC ranks third in term of incidence, and second in terms of mortality [1,2]. The majority of CRCs arise from precursor lesions such as adenoma, which later develop into adenocarcinoma. Adenocarcinomas originating from epithelial cells of the colorectal mucosa account for more than 90% of CRC cases [3,4,5].

A combination of genetic/epigenetic and environmental factors, widely described in the available literature, contribute to the onset and development of CRC [6,7]. However, there are still many open questions regarding, e.g., the initiation of neoplastic lesions [4]. An attractive hypothesis for the initiation and growth of CRC is “the cancer stem cell (CSCs) concept hypothesis”, based on the presence of a small subpopulation of cells with embryonic stem cell (ESC) characteristics. Most research work focuses on the identification of genes involved in the induction and pluripotency of stem cells (SCs) and shows the markers of CRC CSCs (reviewed in [5]). However, a significant difficulty in these studies relates to the fact that CRC cells not only exhibit multi-endocrine features, producing different types of neurohormonal polypeptides, but also amphicrine characteristics (epithelial mucin-positive cells) of different grades. In addition, most of the cells in the intestinal crypts adjacent to the tumor also exhibit the described features [8,9]. There are suggestions of close cooperation between CSCs and neuroendocrine cells (NCs), which reside adjacent to colonic SCs in the crypt SC niche [10,11].

Furthermore, the somatotropin-release inhibitory factor/somatostatin (SRIF/SST) system plays a major role in the maintenance of CSCs in a quiescent state [10,12]. In the central nervous system (CNS), SST serves the role of a typical neurotransmitter/neuromodulator, regulating locomotor and cognitive functions. In turn, in peripheral tissues, mostly the gastrointestinal (GI) tract, it serves as a pan-inhibitory peptide in processes of endocrine and exocrine secretion. It also affects the motility, blood flow, and intestinal absorption, and exhibits anti-inflammatory activities [13,14,15].

Furthermore, the SRIF system is characterized by a strong antiproliferative activity, increasing cell apoptosis and inhibiting angiogenesis in most of the cancerous tissues [12,16,17,18,19,20]. This property is already used in clinical practice [20,21,22,23,24,25,26]. SST acts through the activation of five membrane receptors (SSTRs/SSTs), belonging to the G protein-coupled receptor (GPCR) family [12,16,19,24,27,28].

The most abundant source of SST in the GI tract is a group of highly specialized intestinal endocrine cells, known as enteroendocrine cells (EECs) restricted to the mucosa, termed D cells in the stomach [29,30,31,32,33], as well as delta (δ) cells in Langerhans islets of the pancreas [34,35,36,37]. SST in the GI tract is also produced by neurons of the enteric nervous system (ENS) [24,29,30,36,38,39,40,41]. However, the majority of the circulating SST is secreted by gastrointestinal D cells (~65%), ~30% by the CNS, and ~5% by pancreatic D (δ) cells [37].

SST is a peptide that assumes two forms (SST-14 and SST-28), secreted mostly in a paracrine/neurocrine fashion, released in a pulse manner as a very short-lived peptide of about 3 min bioactive half-life in circulation, where it is degraded rapidly by ubiquitous peptidases [12,22,24,37]. In CNS, the peripheral nervous system (PNS) and pancreas SST-14 is the dominating form, while SST-28 is secreted by D cells in the GI tract [42].

Differential and often very strong expression of SSTRs (mRNA, protein) has been detected in a number of solid tumors in humans [43,44,45,46,47,48,49]. This mostly concerns gastroenteropancreatic neuroendocrine tumors/neoplasms (GEP-NETs/NENs) or mixed adenoneuroendocrine carcinomas (MANECs) [50,51,52]. However, more studies are needed to determine the role of the SRIF system and the mechanism of its action in the pathogenesis of non-NETs in the GI tract, including sporadic CRC [49,53,54]. The remaining questions include, e.g., the role of the SRIF system in histogenesis of colorectal adenocarcinoma cells [8,9,52,55,56], as well as the origin of signet-ring cells in mucinous adenocarcinoma [57] and SC overpopulation [10]. Furthermore, the diagnostic/prognostic role of the tissue expression of SRIF system components in sporadic CRC is still subject to discussion. Similarly, the mechanisms of immune system control via SRIF system in colorectal cancers are also poorly understood [20,58]. In the most recent literature, the participation of the epigenetic mechanisms in the modification of the expression of the SRIF system factors is often discussed, as there is a possibility that the knowledge of such could result in improvements in cancer therapy [49,59]. Methylated SST is one of the biomarkers of early CRC detection [59].

This review summarizes the recent collective knowledge and understanding of the clinical significance of the SRIF system signaling in colorectal cancer, aiming to evaluate the potential role of this system’s components in CRC histogenesis, diagnosis and potential therapy.

## 2. The SRIF System—General Comments

The SRIF system comprises seven genes encoding two peptide precursors, somatostatin (SST/SRIF) and cortistatin (CST), as well as five receptors [12]. The human SST gene contains a singular intron, interrupting the coding sequence in the propeptide region of the molecule [60,61]. This gene is localized on chromosome 3 and has one transcript and 262 orthologs [62]. SST protein has two biologically active forms: 14 amino acid (AA) form (SST-14) and a 28 AA form, derived from the larger precursor preprosomatostatin of ~120 AA [63]. It is not clear whether the two peptides are co-expressed by the same or separate cells [12].

### 2.1. Somatostatin (SST) Localization and Role in Physiology

SST was first described over 50 years ago in rat [64] and sheep hypothalamus extracts [65], and was known since then as the somatotropin release-inhibiting factor/hormone (SRIF, SRIH) or growth hormone release-inhibiting factor/hormone (GHIF, GHIH). In turn, first isolation and characterization of this tetradecapeptide with a sequence of H-Ala-Gly-Cys-Lys-Asn-Phe-Phe-Trp-Lys-Thr-Phe-Trh-Ser-Cys-OH was performed on material derived from sheep hypothalamus [65]. Both of the mentioned studies related to the SST-14 form of this protein, with the N-terminally extended version SST-28 described later [66]. Isolation and sequencing of the gene encoding human SST [60,61] allowed the discovery of the biochemical and biological properties of this peptide, as well as the regulation of its gene expression in cells of different tissues and organs.

Moreover, the presence of SST was detected in the anterior and posterior lobes of the pituitary gland, cerebral cortex, spinal cord, and PNS [12,34]. It was detected in the auditory and visual sensory neurons, and nerves of the sympathetic and parasympathetic nervous system [34,67]. Moreover, a recent paper confirms the broad distribution of SST in the CNS and describes the role of this peptide in local synaptic transmission and in spreading into the extracellular space via volume diffusion (reviewed in [68]).

The ultrastructural characteristic of SST-producing cells in the nervous system is their round shape, electron dense secretory granules, cytoplasmic elongations, flocculent matrix, and closely apposed limiting membrane. Nerves characterized by SST presence contain large, dense P-type neurosecretory granules, distinct from those storing other peptidergic neurotransmitters [69]. In neurons of rat dorsal root ganglia, co-localization of SST with other peptides has been described, e.g., calcitonin gene-related peptide (CGRP) and tachykinins in the same secretory granules, suggesting that SST cooperates with other peptides in neural stimulation at the synapse level [70].

In the GI tract, the majority of SST production occurs in mucosa (>90%), mostly of the stomach, the duodenum, and the jejunum, with <10% taking place in the submucosal and muscle layers. In the mucosa, SST is localized in epithelial endocrine cells/enteroendocrine cells (EECs) [24,29,30,36,38,39,40,41]. The most abundant sources of SST in the GI tract are intestinal EECs, termed δ- or D cells in the antral and the fundic mucosa of the stomach. D cells are described as of a “closed type” in the fundic mucosa, not showing any luminal contact, and “open type” in the antrum, as their apical membrane is fused with the gastric lumen. SST regulates intragastric pH via restriction of acid secretion from parietal cells [29,30,31,33], inhibiting gastrin production in the G cells and histamine secretion from enterochromaffin-like (ECL) cells [33], as well as delaying gastric emptying [32]. SST-producing cells also include δ-cells, constituting around 5% of endocrine cells in Langerhans islets in the pancreas [34,35,36,37]. Pancreatic δ-cells are neuron-like, forming a network for intra-islet communication. In humans, they are scattered throughout the islets and are intermingled with α- and β-cells, with a relatively small number of axon-like projections [71]. Morphological descriptions of these SST-producing cells are available in the literature [41,69,72]. Negative feedback between δ- and β-cells in the pancreas affects glycemic control, and any disruption of these interactions plays a role in the pathogenesis of diabetes [41,71]. In recent years, the possibility of converting non-β islet cells (including δ-cells) into β-cells to replenish β-cell mass has also been explored as a way to treat diabetes [73]. Apart from D cells in the GI tract, SST can also be produced by other EECs, known as the K cells in the jejunum, in which it is co-expressed with glucagon-like peptide 1 (GLP-1), secretin, cholecystokinin (CCK), and peptide YY (PYY) [32,74,75,76]. The weaknesses of letter classification are highlighted when it comes to EECs (e.g., D and K cells) that produce SST in the GI tract. According to some authors, alternative names should include the site of hormone secretion, species, and detected peptides/hormones [75,76].

In one of the more recent taxonomies of small intestinal cell subtypes, SST was specific for the cellular SAKD subset [77]. The cells that produce SST in the intestine (K cells) are “flask-shaped”, with apical membranes exposed to the lumen [75]. In mice, K-cells located in the upper small intestine were described to overlap in 10% with GLP-1 and in 6% with SST [74]. However, the intestinal SST, as opposed to stomach SST, is produced mostly in ENS structures of the submucosal and myenteric plexuses (in Dogiel type II neurons) [38,39,40]. In the human colon, similarly to the small intestine, co-localization of SST and calretinin (CALR) indicates type II neurons as a primary source of SST. However, around half of SST(+)/CALR(−) cells was also made up of neurofilament-reactive, multi-axonal type II neurons [40].

While SST expression was also detected in rectal mucosa, it was present in a smaller number of endocrine cells compared to bovine pancreatic polypeptide (BPP)-, human PP-, and glucagon-like immunoreactive cells [78]. In turn, other authors consider SST as one of the four most common hormones produced in EECs, next to 5-hydroxytryptamine (5-HT, serotonin), GLP-1, and PYY. Furthermore, among the 5-HT cells, the most common cell type in normal human colon, some also exhibited co-expression of SST and PYY, more rarely GLP-1 [79]. The cells producing 5-HT, as the most widespread EECs of the GI tract, are often referred to as enterochromaffin cells (ECs). They are also among the most common subtypes in the colon and rectum [80]. Moreover, there is currently some proof of a significant heterogeneity of these cells, characterized by co-localization of 5-HT and several peptide hormones [81].

Further locations of SRIF system component production include endocrine and non-endocrine cells, such as thyroid C cells [82,83,84,85,86], granulosa cells of the corpus luteum, tubular and Leydig cells of the testis, adrenal gland cells of the cortex and medulla [86], human tubular epithelial cells, and glomerular cells in the kidney [86,87].

SST produced in peripheral tissues has no relation to GH secretion but exhibits a superior function (mainly inhibitory) in the regulation of the secretion of hormones of the GI tract and pancreas (e.g., gastrin, secretin, CCK, gastric inhibitory polypeptide (GIP), vasoactive intestinal peptide (VIP), enteroglucagon, motilin, insulin, and glucagon). It regulates gastric acid, digestive enzymes, bile, and colonic fluid [12,37,41,88]. Furthermore, it suppresses gallbladder contraction, small intestinal segmentation, and gastric emptying [12,32]. In the pancreas, SST acts locally, within the islets, as a paracrine inhibitor of insulin and glucagon secretion [12,37,41]. In the model of mouse pancreatic islets, a cycle of mutual feedback loop in the expression of SST and glucagon was described. Thus, SST inhibits glucagon secretion at low and high glucose levels, and glucagon stimulates SST secretion via glucagon and GLP-1 receptors. Therefore, glucagon is essential for normal physiological SST secretion. Additionally, it was demonstrated that while SST strongly inhibits insulin secretion, insulin has no direct effect on the secretion of SST in mouse islets [89]. During paracrine action, SST is an inhibitor of other EECs and excitatory cells, as well as affects colonic motility [88,90,91]. It also exhibits local immunomodulatory action in the gut [75].

While autocrine effects of SSTs are also reported in the literature, the detailed mechanisms are still unclear [28]. Furthermore, SST can be secreted directly into the GI tract lumen [31,37,67,91].

### 2.2. SST Secretion Control in the Gastrointestinal Tract

Although the sites of SST secretion and release in physiology are well described, their mechanisms are less understood (reviewed in [28]). SST is produced as a precursor preprosomatostatin, in rats and mice processed enzymatically at its C-terminal site in the secretory granules by a proprotein convertase (PC), most probably PC2 [92], to yield several mature products. Three widely distributed peptides, namely SST-14, SST-28, and SST-28(1-12), are derived from prosomatostatin, more specifically from its C-terminal precursor. In the antrum of rat stomach, an additional decapeptide derived from the NH2 end of the prohormone was identified and named as antrin. It was described to be present in the secretive granules of the pyloric D cells, together with SST-28(1-12), both in humans and animals [93]. Human prosomatostatin is a protein build from 92 AAs, and is proteolytically cleaved at the arginine–lysine and arginine processing sites. Partial micro-sequencing of prosomatostatin-(1-63)-peptide demonstrated that the site of the cleavage of the signal peptide was located at the Gly24-Ala25 bond [94].

Regulation of SST secretion in the GEP system is affected by three groups of factors: (1) intraluminal, (2) neural, and (3) endocrine. The first group includes nutritional factors and acid (low pH) in the stomach, and duodenum, which promote the antral D cell to secrete SST (the negative feedback regulatory loop). However, the control of gastric SST secretion is influenced by both nutritional components and stimulation of the vagus nerve and ENS structures with the neurotransmitters/peptides it secretes [31,73].

Release of gastric SST via a neural mechanism was linked to an opposite activity on the sympathetic (stimulates) and parasympathetic nervous system (inhibits). Locally secreted neurotransmitters/peptides include, e.g., VIP, CGRP, and pituitary adenylate cyclase-activating peptide (PACAP). Moreover, a stimulating activity of incretin hormones, cholecystokinin (CCK), acetylcholine, oligopeptides, and amines was demonstrated on SST secretion by stomach D cells [31]. In the same study model, the stimulatory effect of ghrelin was demonstrated on SST secretion by pancreatic islet δ cells [95]. Glucose stimulated SST expression by the δ-cells of the pancreas is also promoted by urocortin3, produced by β-cells [96]. Furthermore, the process of SST expression in human pancreas cells occurs in a Ca^2+^-dependent manner [97].

In turn, endocrine factors that stimulate SST release in GI tract include hormones such as gastrin, CCK, bombesin/gastrin-releasing peptide (GRP), oxyntomodulin, GLP-1, and CGRP, while inhibiting factors consist of, e.g., substance P, insulin, glucagon, and pancreatic polypeptide. According to the study of Salehi et al. conducted on rat pancreas, the release of SST is slightly delayed in relation to insulin, and anti-synchronous to glucagon [98]. Pulse secretion of all three hormones occurs due to the activation of P2Y(1) receptors and is modulated by extracellular ATP. It was also proven that the A(1)R adenosine receptor is important both for the amplitude (insulin) and duration (glucagon and SST) of the pulse secretion of hormones from the Langerhans islets [99].

In contrast, the molecular mechanisms of regulation of SST secretion particularly emphasize the role of enhancers and silencers in the promoter region of the gene, as well as the binding of modulatory transcription factors to these elements. The roles of pre-translational mechanisms regulating the expression of this peptide (e.g., methylations and polymorphisms within the promoter region, activity of various transcription factors) and post-translational mechanisms (e.g., proteolytic cleavage of preprosomatostatin to SST-14 and SST-28, peptide secretion) were also confirmed [18,28]. However, more studies are needed to fully elucidate other mechanisms of SST regulation (e.g., miRNAs, alternative splicing, autocrine feedback, and protein modification) [28].

### 2.3. Somatostatin Receptor Localization and Role in Physiology

All five somatostatin receptors (SSTRs) with seven transmembrane-spanning domains are prototypical class A GPCRs that belong to the rhodopsin-like family of receptors (reviewed in [12]). Human SSTRs are encoded on five different chromosomes and only one of the receptors undergoes alternative splicing (SST2), generating two isoforms known as SST2A and SST2B [16,27,100,101]. SSTRs have affinity for both forms of SSTs (SST-14 i SST-28) [27,100,102,103,104]. Each receptor subtype is coupled to different signal transduction pathways through G-protein-dependent and -independent mechanisms [12,27,100]. The receptors share common pathways involving G-protein-dependent mechanisms of adenylate cyclase (ACL) inhibition, activation of protein phosphotyrosine phosphatases (PTPs), and modulation of mitogen-activated protein kinase (MAPK, ERK) [100,105]. The SRIF signal pathways are well described in the literature [16], prompting the author to attempt to provide a different scope of information. However, it is worth highlighting that the main pathways regulated by the activation of SST1–5 lead to the inhibition of secretion (e.g., other neuropeptides, hormones, growth factors, and cytokines), cell proliferation, migration, and angiogenesis [12,16,27,100,106,107] (Table 1).

SST1–5 localized on cell membranes are connected to transmembrane K^+^ channels (also known as G-protein gated inwardly rectifying potassium channel (GIRK1), Ca^2+^ channels (also known as voltage-operated calcium channel, VOCC), as well as intracellular enzymes, mostly ACL and PTPs. Among PTPs, the most important intracellular SSTR effectors are Src homology 2 (SH2)-containing tyrosine phosphatase-1 (SHP-1), SHP-2, and phosphotyrosine phosphatase DE-1/PTPeta [12,105]. Upon binding to SST, intracellular pathways are activated by SST1–5, leading to antiproliferative and anti-secretory effects. In addition, activation of SST2 and SST3 also exerts pro-apoptotic effects [106,107]. Recent studies on human brain extracts indicate that SST binds primarily to several members of the P-type ATPase family. Subsequent validation experiments confirmed the interaction between SST and the sodium–potassium pump (Na^+^/K^+^-ATPase) and identified the tryptophan group in SST as a critical component of the binding. Functional analyses in three different cell lines showed that SST can negatively modulate the rate of K^+^ uptake by the Na^+^/K^+^-ATPase [108].

A characteristic property of SSTRs is also absolute or relative subtype selectivity, related to its inhibiting influence on various biological processes, e.g., hormone and mediator secretion, exocrine secretion, motility, and cell proliferation [12,19,27]. As an example, inhibitory effect on GH secretion occurs mainly via SST1, SST2, and SST5; insulin secretion via SST2 and SST5; glucagon secretion via SST2; and cytokines, i.e., interleukin 6 (IL-6) and interferon γ (IFN-γ), histamine and immune responses, through SST2 [27,100]. In turn, the inhibitory effect on colonic contraction occurs mainly through the activation of SST5 [27].

SST1–5 expression of varying intensity is demonstrated in numerous normal human tissues and organs, encompassing both NCs and many non-neuronal and non-endocrine cells [24,27,86,100,109,110,111]. Furthermore, many immune cells (e.g., monocytes/macrophages, B and T lymphocytes, and dendritic cells) show expression of SSTRs [14,110,112]. The most detected subtype is SST2, with a mostly membranous immunohistochemical (IHC) reaction, while SST4 is the least functionally described [12].

SST-14 binds with higher affinity to SST1-4, whereas SST-28 mainly interacts with SST5 [16,100]. Two subfamilies of SST receptors have been described on the basis of chemical structure identity and pharmacological characteristics: the first class, comprising SST2, SST3, and SST5, binds synthetic SST analogues (SAAs), such as octreotide (OCT) and lanreotide, while the second-class receptors, SST1 and SST4, do not interact with these agonists [113]. The basic signaling pathways in the action of SST are also known, ultimately exerting anti-secretory, anti-proliferative, and pro-apoptotic effects (via SST2 and SST3) [12,107].

Upon binding, SSTRs are phosphorylated, internalized into clathrin-coated vesicles, and directed to endosomes. Receptors can then be directly returned to the cell membrane or targeted by the proteasome pathway [101,114]. The particular receptor types differ in internalization rates (internalization is higher for SST2, SST3, and SST5 than for SST1 and SST4). This property affects the fates of the receptors, with SST2 and SST5 recycled more rapidly, while SST3 is commonly degraded [114].

The role of the SRIF system in the GI tract is widely described in a recent review [24]. The presence of SSTRs has a potential role in cancer pathogenesis (including CRC) [52,54,115,116,117,118,119,120], as well as in non-somatostatin receptor-related diseases (e.g., inflammation, the granulomatous diseases; reviewed in [111]). Inhibition of cell proliferation in tumor cells occurs with the participation of all SSTRs.

## 3. The SRIF System and Different Tumors

Apart from normal tissues, the expression of SRIF system components concerns a range of human cancers [16,17,19,48,53], mainly NENs/NETs of GI tract [45,107,121,122,123]. SST-producing NETs are derived mostly from the pancreas, duodenum, and small intestine [124,125,126]. Most of well-differentiated gastroenteropancreatic endocrine carcinomas (WDEC) also demonstrate the presence of SSTRs, mainly SST2 (including SST2A) and SST5 [121,122,126,127]. However, the most prominent SST subtype, detected in GEP-NETs, is SST2/SST2A, identified in >70% of cases at a high expression intensity [45,47,121,122,128,129]. The occurrence of a high density of SSTRs has prognostic significance and may be used in the therapy of these cancers [45,47,126,127,129].

Production of SST and SSTRs was also detected in thyroid medullary carcinomas, both in vivo and in vitro [130,131,132,133], pituitary adenomas [134,135], as well as tumors of adrenal glands [136] and ovaries [137]. SSTR expression was also described in non-endocrine neoplasms, such as breast cancer [138] or pancreatic ductal adenocarcinoma (PDAC) [139]. In breast cancer, the dependence of SST expression on the status of estrogen and progesterone receptors was observed [138]. In the case of PDAC, it was shown that the methylation of *SST* promoter is a sensitive and promising molecular biomarker of this cancer [139].

## 4. The SRIF System and Large Intestine in Physiology

Although the main source of SST in the GI tract is the stomach [31], using various assay techniques, SST expression was also demonstrated in the human colon [30]. In the latter location, it mainly occurs in ENS structures, which include neurons of the intramuscular plexuses, external and internal, submucosal plexuses, intra-ganglionic nerve fibers and nerve fibers in mucosa and muscularis propria (reviewed in [39]). SST is counted as one of three neuropeptides also produced by EECs in the human colon alongside glucagon and BPP [11,80]. According to the proper nomenclature, these cells are termed D cells [88,91], which make up ~3–5% of the EEC population and form the seventh cluster of the EEC population in the lower GI tract in humans and various animal species [91]. In IHC analysis of normal colonic crypts for neuroendocrine markers, the highest number of SST-positive cells was detected in the bottom half crypt region. However, the proportion of these cells in the total number of crypt epithelial cells is very low, lower than chromogranin-positive cells. Moreover, a similarly low amount of SST receptor type 1 (SST1) is observed in this location [11].

In the case of normal human colon epithelial cells, IHC positivity was demonstrated for all SSTRs except SST4. In contrast, normal rectum epithelial cells showed the presence of only SST1 and both SST2 subtypes (A and B), while SST3, SST4, and SST5 were not detected [110].

Little work has addressed the role of the SRIF system in physiological colorectal function. In this regard, mainly the inhibitory influence of SST is described via SST5 on colonic contraction [12,27]. However, studies of the effect of this peptide on the spinal defecation center in an animal model (rat) showed the opposite effect (enhancement of colonic motility). This could explain the simultaneous occurrence of chronic abdominal pain and colonic motility disorders in IBS patients [140]. Interestingly, in the context of SST function in normal human colon, there are descriptions of SST-immunoreactive fibers on submucosal, but not mesenteric vessels, which suggests the role of this peptide in the control of blood flow to the human gut [141]. In rats, SST2 was shown to mediate the anti-secretory effects of SST in colonocytes [142]. In an in vitro model (HT-29cl.19A colonic cells), inhibition of chloride secretion by SST was described [143]. The motor and sensory effects of SST in the colon are likely mediated by SST1 and SST2, with SST1 and SST2 localized in the longitudinal and circular colonic smooth muscle, respectively [144].

Recent in vivo (mice) and in vitro studies (human goblet-like cell line LS174T) with administration of exogenous SST (octreotide, OCT) indicate that SST could promote the expression of mucin 2 (MUC2) and secretion of mucus by these cells. This action occurs through binding of SST to SST5 and suppression of Notch-Hes1 signaling [145].

## 5. The SRIF System and Inflammatory Bowel Diseases

SST, as an inflammatory inhibitory peptide, has potential importance in the pathogenesis of inflammatory bowel diseases (IBDs) (reviewed in [146,147]). These conditions, which include ulcerative colitis (UC), Crohn’s disease (CD), and microscopic colitis, are risk factors for ~3% of CRC [148]. The involvement of multiple neuropeptides (including SST) and mechanisms of action in pre-cancerous alterations and colonic inflammation were described in an earlier publication [149]. On one hand, correlations are observed between the expression of neuropeptides from GI tract and changes in immune cells during inflammatory process [146,147]; on the other hand, special attention is paid to the interactions between GI tract neuropeptides/amines and gut microbiota, a crucial component in the pathophysiology of IBD [20,147]. The anti-inflammatory effects of SST include stimulation of B-lymphoblast proliferation with increased immunoglobulin production [15], inhibition of T-lymphocyte and granulocyte proliferation, and reduction of proinflammatory cytokines, such as IFN-γ [13,14,30].

Because the GI tract is a major source of the circulating form of SST, as early as the 1990s, plasma SST levels in patients with various GI tract diseases were studied. These levels ranged between 46 and 73 pg/mL. A post-prandial increase in SST was observed in all patients but was significantly higher in patients with duodenal ulcers (159 ± 20 pg/mL), active UC (176 ± 17 pg/mL), and irritable bowel syndrome (194 ± 20 pg/mL). Moreover, the postprandial elevation of plasma SST concentrations was suggested to be influenced by vagotomy, with the particular increased elevation occurring due to gastric hyperacidity, acute lesions of the colonic mucosa, and hypermotility of the GI tract [150].

Relatively little is known about the role of tissue expression of the SRIF system components in IBD pathogenesis. There are differences regarding SST expression in IBD patients and animal models of colitis [151,152,153]. The most often used models of colitis include dextran sodium sulfate (DSS)-induced colitis and trinitrobenzene sulfonic acid (TNBS)-induced colitis, which can mimic human IBD, including UC [154] or CD, respectively [155]. In IBD patients, the main form of SST detected using specific radioimmunoassay was SST-28. A fall in expression of this peptide was observed in mucosa submucosa and muscularis externa layers in UC and in CD vs. normal colon. The reduction in SST detection was greater in more severe forms of colitis compared to minimal lesions, possibly due to a decrease in EECs with the severity of inflammatory lesions [151]. Similarly, in another study, the number of SST-secreting EECs was reduced in IBD patients compared with controls. This reduction was related to the level of inflammation in CD—the higher the degree of inflammation, the lower the number of SST-positive cells [156]. Interestingly, a reduction in SST-immunoreactive nerve fibers was also observed in IBD. Changes in perivascular nerves may account for the congestion and ulceration characteristic of IBD. Moreover, neural changes may be the underlying source of pain [157].

When it comes to animal colitis models, an increase in the number of SST-producing cells in the colon tissues was described in TNBS-induced colitis compared with healthy controls. Furthermore, the density of SST-immunoreactive cells positively correlated with the number of macrophages/monocytes and mast cells [152]. In turn, in DSS-induced colitis in rats, a lower density of PP- and SST-positive cells was detected compared to control group, similarly to the results of IBD patients. These changes in all EECs were accompanied by an increase in the densities of mucosal leukocytes, T and B lymphocytes, macrophages/monocytes, and mast cells vs. control. Regardless of the IBD model, these studies support the theory of a cooperative relationship between EEC-produced peptides (including SST) and immune cells in IBD [153].

Nevertheless, the results of studies on tissue expression of SSTRs in IBD are sparse and concern practically only SST2 [158,159]. A predominantly membranous pattern of IHC reaction was detected on NCs. The number of SST2-positive cells was significantly lower in patients with CD than those with unchanged mucosa and unclassified colitis. Cells with cytoplasmic expression comprised intraepithelial T cells, while cells with membrane cytoplasmic expression included both NCs and epithelial cells [158]. Moreover, in an available case report of a CD patient, SST2A expression was observed in a pathological lesion composed of a fibrillar network of fine nerves, extending deep into the inflamed area of the subepithelial portion of the mucosa [159].

## 6. The SRIF System and Colorectal Cancer

The vast majority of histological CRC subtypes are adenocarcinomas [3,4,5], among which moderately differentiated adenocarcinomas are the most prevalent (~70%). Well and poorly differentiated carcinomas account for 10% and 20%, respectively. Furthermore, other rare types of CRC include neuroendocrine, squamous cell, adenosquamous, spindle cell, and undifferentiated carcinomas [3].

### 6.1. Somatostatin Tissue Expression In Vivo

Studies regarding tissue expression of SST in CRC are few and inconsistent. Moreover, only some of them involve simultaneous examination of cancer tissue and normal colonic mucosa from the same patient (control). Immunoreactive structures include cancer cells, EECs, tumor-neighboring colon crypt cells [9,18,55,160,161], as well as neurons and nerves present in the colonic ENS [162]. In the case of NETs of the rectum (*n* = 32), SST was the most commonly detected neuropeptide (in 35% of tumors) [163]. In turn, another study of large intestine carcinoids (*n* = 84), SST-positive IHC reaction showed only in 3% of tumors [164].

SST expression of varying proportions is described in normal human colon mucosa [8,9,10,18], as well as sporadic CRC tissues [8,9,55,160,161,162,165]. In these studies, the frequency of detection of SST in tumor and control tissues was similar [8,9,162] or reduced in CRC vs. control [18]. Moreover, in two studies based on mRNA assessment of SST (qRT-PCR), levels of this protein were reduced in CRC compared to control [18], or not detected at all [10]. Negative reactivity to SST (but positive for bombesin and VIP) in a patient with signet-ring cells adenocarcinoma was also demonstrated [57] (Table 2).

Regarding the prognostic aspect of SST expression in CRC, only one study highlighted lower levels of this neuropeptide that correlated with grading, with lower SST expression detected in poorly differentiated tumors. The relationship between SST expression and CRC metastatic potential was not investigated [161].

In vivo SST expression studies show that most SST-immunoreactive cells were a component of the altered glandular structures of CRC. The mention of D cells within the colon also appears sporadically in literature that used electron microscopy to evaluate subpopulations of immunopositive cells [161]. Some of the studies demonstrated co-localization of SST with other GI tract endocrine neuropeptides/markers (e.g., chromogranin A, serotonin, glucagon) within single cells [55,160].

In summary, the above results are difficult to comment on due to the heterogeneity of the tissues of the CRC itself, significant differences in the amount of tissue tested from patients, the absence of control groups in some studies, and methodological differences (e.g., type and dilutions of anti-SST antibodies, scales for quantifying IHC reactions). In general, the presence of SST in CRC cells from corresponding fragments of the normal GI tract and other GI tract adenocarcinomas was confirmed [24,29,30,36,39,41]. However, SST expression appears to decrease with tumor grading and is lower in poorly differentiated CRC or mucinous subtype of CRC with signet-ring cells.

In turn, our studies, based on immunohistochemistry assays, demonstrated individual scattered SST-immunoreactive cells located between the epithelial cells lining the normal colon mucosa. The latter were EECs, morphologically similar to serotonin-positive cells (ECs) (unpublished data) (Figure 1A,B). In contrast, CRC tissues showed multiple cancer cells with a cytoplasmic pattern of IHC reaction distributed throughout the cell or a granular IHC reaction predominant in the apical portion of the cancer cells (Figure 1C,D).

### 6.2. Somatostatin Receptor Tissue Expression In Vivo

SSTR presence (mRNA, peptide) was investigated in neuroendocrine tumors, MANECs, and pure colorectal adenocarcinoma (Table 2). This expression affects both normal colon mucosa and primary and/or metastatic tumor tissues, regardless of histologic subtype. The cells with positive expression included both normal absorptive cells (colonocytes) and tumor cells. SSTR expression was also described in immune cells in lamina propria and tumor-adjacent stroma. Most studies attempted to simultaneously study all five subtypes of SSTRs. However, there are large differences in the frequency of individual SSTR expression and its intensity in literature sources.

In the case of SST1, heterogenous [166] and quantitatively differentiated expression of this receptor was described, from rare [10,167] to common [115] to even dominating receptor type in CRC and control tissues [118]. Only one group of authors described a positive correlation of this receptor’s expression with tumor stage and lymph node metastasis [118].

Observations on tissue expression of SST2 are also varied. However, most authors described a frequently detected type of SSTR both in CRC and control samples [115,117,118,138,166,167]. SST2 production in the tumor correlated, according to some observations, with tumor staging [166], tumor type [118], localization [119], or serum concentration of carcinoembryonic antigen (CEA) [117,138]. Lower or no SST2 expression was described at higher tumor stages (Dukes’ C, D) [166], ulcerative CRC type [118], or increased patient CEA levels [117,138]. Higher expression was also noted in tumors located in the rectum [119]. Raggi et al. observed a correlation between higher SST2 mRNA expression and an increase in cancer-related death and shorter disease-free survival (DFS) [54]. In turn, Evangelou et al. reported opposite findings. They observed that while SST2 (and SST5) expression is a good prognostic when it comes to survival rates, they were not independent predictors of survival [119].

In contrast to the lack of SST2A expression in colorectal MANEC and poorly differentiated NETs [52], positive IHC reaction for SST2 was described in a patient with poorly differentiated anal neuroendocrine carcinoma (ANECs) with regional lymph node metastases [168].

SST3 and SST4 expression in CRC and normal mucosa were generally low or rarely detected. In one of the papers, quantitative differences were described in the frequency of SST4 expression in tumor cells (more) vs. normal mucosa cells (less) (Table 2). In the case of SST3, a correlation of this receptor’s expression with stage was observed, with similar dependency described between SST4 and grade [118].

In the case of SST5, the frequency of this receptor’s expression in CRC and control tissues was generally high, sometimes occurring more frequently in CRC vs. control tissues [116]. This expression was indicated as dominating among other receptors [115] or “second” SSTR subtype [118]. Furthermore, both a decrease in expression with an increase in stage [118,167] and lack of such correlation were described in the literature [115]. It was observed that while SST5 (and SST2) expression could be good prognostics when it comes to survival rates, they were not independent predictors of survival [119].

Interesting studies conducted in submucosal and subserosal vessels showed 3–5-fold overexpression of SSTRs and substance P receptors in the host veins within a close area (2 cm wide) surrounding the human CRC, as compared with veins located at a greater distance (5–10 cm) in control tissue. These results suggest the presence of a regulatory mechanism in the tumor vascular bed, which could be key for the development of CRC metastasis mechanisms [109].

In conclusion, studies on the expression of SSTRs in pure CRC adenocarcinoma are characterized by the great heterogeneity of the results obtained. However, it appears that the most abundantly represented subtypes of SSTRs in this tumor are SST2 and SST5. The highest correlation with clinical data was shown for SST2, followed by SST5. As for the prognostic value of receptor expression, however, the results are surprisingly divergent and also mainly involve SST2 and SST5.

Own study showed membranous expression of SST2, and cytoplasmic pattern of positive IHC reaction for SST3 and SST5 in normal colon and colorectal cancer tissue samples from the same patient. More numerous immunoreactive cells were observed in CRC as compared with adjacent normal mucosa (unpublished data) (Figure 2).

A summary of data regarding cellular sources of somatostatin (SST) and SST1–5 in colorectal cancer (CRC) indicates that effector cells of the SRIF signaling system are specific for various SSTRs (Figure 3). As mentioned, the local effects (paracrine, autocrine) of endogenous SST mostly involve inhibition of cell proliferation versus promotion of apoptosis, and inhibition of production and secretion of many other tumor growth factors. All mechanisms leading to reduced or absent SST expression in CRC in vivo may result in impaired anti-tumor effects of this neuropeptide.

The potential role of SRIF system components produced by immune cells has been studied in pathogenesis of IBDs (as noted in Section 5). In patients with CRC, expression of some SSTRs (e.g., SST1, SST2, and SST4) on immune cells was confirmed in tumor stroma [116], but the exact role of SRIF system components has not yet been fully elucidated and requires further research.

Epigenetic alterations (mainly SST methylations) present in a high percentage of CRC contribute to a decrease in expression and impaired protective function of SST in CRC, as is described later.

### 6.3. The SRIF System Component Expression in Colorectal Cancer In Vitro

CRC cell line studies (HT-29, Caco-2, HCT-15, HCT-116, SW480 cells) demonstrated differential expression of SST and SSTRs [10,11,116,169,170,171]. Vuaroqueaux et al. detected major expression of the SST5 protein in the HT29-D4 cell line [116]. In turn, Hohla et al. demonstrated the expression of mRNA encoding SSTRs and high affinity binding sites for SST in all three of the analyzed cell lines (HT-29, HCT-15, and HCT-116) [170]. Furthermore, an increase in SST1, SST2, and SST5 mRNA expression was observed in Caco-2 and HT-29 cells. Additionally, the expression of both SST2 isoforms (A and B) and SST4 on cell membranes of these cell lines was demonstrated using the IHC method [171]. Other authors observed that both HT-29 and SW480 cells express SST, SST1, SST2, and SST4, but the transcript level is more abundant in HT29 than SW480 cells [10].

Although few in number, in vitro studies have firstly demonstrated the antiproliferative effect of SST and its synthetic analogue (OCT), and revealed some mechanisms of this effect on CRC cells [169,171,172,173,174]. Thus, HT29 cell growth was inhibited by SST-14 only in the presence of serum, with a maximal and significant response at a concentration of 10^(−7)^ M [172]. In turn, Colucci et al. demonstrated that SST-14 inhibited basal cyclooxygenase-2 (COX-2) expression, prostaglandin E(2) (PGE(2)) production, DNA synthesis, and growth in Caco-2 cells. Downregulation of COX-2 expression and function in CRC cells by SST is thought to occur through activation of SST3 or SST5, and these effects contribute to the antiproliferative effects of SST on tumor cells [169]. In a SW480 cell model, it was demonstrated that OCT inhibited growth, induced apoptosis, and arrested the G1 cell cycle of colon cancer cells in a dose-dependent manner. It was also proven that OCT inhibits human colonic cancer cell growth through the inhibition of the Wnt/beta-catenin signaling pathway [173]. In the same in vitro model (SW480 cells), OCT increased cell apoptosis through SST2 and SST5 activation, promoted β-catenin accumulation in plasmalemma, inactivated T-cell factor-dependent transcription, and downregulated Wnt target genes (e.g., cyclin D1 and c-Myc). Moreover, OCT treatment mediated the activation of glycogen synthase kinase 3 (GSK-3) [174]. In in vitro studies, a potential role of cells exhibiting the expression of selected SSTRs (e.g., SST1) was indicated in the process of inhibition of SC maturation into NCs and overpopulation of SCs. This dysregulation of NC maturation was suggested to be caused by sequential inactivation of adenomatous polyposis coli (APC) alleles in human colonic crypts with SC niche, in which SCs mature into NCs [11].

## 7. The SRIF System and Colorectal Cancer Histogenesis

### 7.1. New Nomenclature of Neuroendocrine Neoplasms

The neuroendocrine neoplasms (NENs) of the GEP system are a histologically, biologically, and clinically heterogenous group. They are derived from the endocrine organs, NCs, and dispersed ECs of GI tract and respiratory system. These tumors usually lack necrosis and are built up of cells rich in cytoplasmic secretory granules and filled with neuroendocrine markers. They produce, store, and secrete biogenic amines and peptide hormones, including SRIF system components [175,176,177]. Poorly differentiated neuroendocrine carcinomas have a particularly poor prognosis [123,176,178].

The most common NENs among this entire heterogeneous group of tumors are GEP-NETs (70% of all neuroendocrine tumors, 2% of all GI tract tumors) [179,180,181]. While the overall diagnosis frequency of GEP-NETs is relatively low, they are still the second most prevalent GI malignancy after CRC in the USA [180,182]. Moreover, the mechanisms of how NE differentiation affects the prognosis of these cancers are still an open question [183]. The classification of NENs originally included (1) well-differentiated NENs, referred to as neuroendocrine tumors (NETs; grade (G) 1 (Ki-67 < 2%) and G2 (Ki-67 2–20%)), and (2) poorly differentiated NENs, referred to as neuroendocrine carcinomas (NECs, G3) (Ki-67 > 20%) [177,184]. G3 tumors were further subdivided into well-differentiated NET G3 and NECs [185].

The term “carcinoid” (or “karzinoide”, introduced in 1907) has often been used to describe GI tract tumors originating from NCs forming the so-called dispersed endocrine system (DES). These tumors were less aggressive compared to carcinomas. The current classification does not prohibit the use of the term “carcinoid”, but indicates it as a synonym for highly differentiated NET. In turn, the term “atypical carcinoid” can be used for NEC with a high degree of differentiation [180,181].

In the current WHO classification (2019), NECs are all considered high-grade tumors. Separate subgroups of NET G3 and NECs are not anymore distinguished because they are genetically unrelated. There are also so-called mixed neuroendocrine–non-neuroendocrine neoplasms (MiNENs) or MANECs (old terminology), in which each component is arbitrarily assumed to account for at least 30% of the tumor [123,186,187,188]. MiNENs rarely contain a well-differentiated NET component in association with a non-neuroendocrine component [189].

### 7.2. Enteroendocrine Cells in Colorectal Cancer Histogenesis

Stem cells are among the intestinal crypt epithelial cells involved in the initiation of carcinogenesis, with their phenotypic characteristics in human colorectal cancer still under investigation [5,190]. In normal colonic crypts, SCs at the crypt bottom generate rapidly proliferating cells, which undergo differentiation during migration up the crypt [191,192]. The normal intestine shows a low number of SCs in the S phase of the cell cycle. During the development of CRC, it was first shown that it is the increase in the number of these cells, rather than changes in the rate of cell cycle proliferation, differentiation, or apoptosis of non-SCs populations, that are important in the initiation of carcinogenesis through an increase in the labeling index rate, i.e., the distribution of crypt cells in S phase in familial adenomatous polyposis (FAP) patients [191]. Furthermore, dysregulation of mechanisms controlling proliferative fraction and S phase probability is thought to explain how germline APC mutations in FAP patients induce an increase in SC population at the bottom of the crypt, shift the population of rapidly proliferating cells upward, and initiate tumorigenesis [192].

Other cells important in histogenesis, particularly of colorectal neuroendocrine neoplasms (C-NENs), are EECs (NC or APUD/DNES cells), which reside adjacent to colonic SCs in the crypt SC niche [10]. There are approximately 17 different types of NCs in the GEP system, but neither the precursor cell nor the biological basis of GEP-NETs are fully understood [182]. In non-neoplastic and neoplastic GI tract tissues and ENS structures, NCs express a panel of identical antigens that are used as neuroendocrine markers. Their presence, even without hormone production, is sufficient to reveal neuroendocrine differentiation [193].

In sporadic CRC, NCs have been identified in a few to >77% of cases (reviewed in [193]). Studies indicate that neuroendocrine differentiation is frequently observed in small cell undifferentiated CRC, which correlates with more aggressive disease progression [194]. Neuroendocrine differentiation is also present in >50% of cases of hereditary non-polyposis CRC (HNPCC) [183]. More NCs also occur in metastatic CRC than in the primary tumor [195]. Furthermore, an increase in the number of NCs is also observed after chemotherapy and radiotherapy treatment [195,196]. The number of tumor cells immunopositive for typical NE markers (chromogranin A, synaptophysin, and CD56) also appears to depend on anatomical location (higher in the right than in the left sided colon) and is similar to the preferred sites of NECs themselves. In normal colonic mucosa, more chromogranin A- and synaptophysin-positive cells were observed in the rectum and the left-sided colon than in the right-sided colon. Hence, the authors suggest that NEC may arise from preceding adenocarcinoma [56]. Previous studies indicate that adenoneuroendocrine carcinomas and neuroendocrine carcinomas are genetically closely related to colorectal adenocarcinomas, suggesting their common intestinal origin [52].

NENs of the colon and rectum have been the subject of intense research, especially in recent years. However, because of differences in epidemiology, prognosis, and treatment principles, tumors in the colon and rectum are usually considered separately [123,176,186,197,198]. Neuroendocrine tumors of the rectum have been diagnosed more frequently since the introduction of screening colonoscopy in the year 2000 [179]. A large US study involving over 13,000 NETs found that one of the three most common locations of carcinoid tumors in the GI tract was the rectum (>27%). Furthermore, 13% of patients affected with such tumors develop a secondary malignancy. The associated non-carcinoid tumors were frequent in conjunction with 20% colonic carcinoids [199]. The most recent works report that colonic and rectal NETs in the USA occur in 0.2 and 1.2 new cases per 100,000 persons/year, respectively [197]. Moreover, pure NECs in the large intestine are diagnosed relatively rarely (0.6–3.9% of all CRC). However, NCs can form mixed tumor fragments with colorectal adenocarcinoma, reaching up to 30% of the entire tumor cell population [52,56,200].

The histologic features of colorectal NETs and NECs are similar to those found in other organs. Colorectal MiNENs consist of a poorly differentiated component and an adenocarcinoma component. Moreover, MiNENs with a low-grade NET component can rarely occur in the background of idiopathic inflammation [181,201]. In turn, when it comes to colorectal NECs, their molecular features are more similar to those of adenocarcinoma than NET [202]. A significantly higher prevalence of abnormal p53 expression (88%), nuclear β-catenin expression (48%), and high cyclin E expression (84%) was described in NEC compared to NET (0%, 5%, and 5%, respectively). The IHC findings of NECs and poorly differentiated adenocarcinoma were similar [202]. In a recent retrospective study from five European Institutions (United Kingdom, France), MiNEN was most commonly present in the large intestine (more than 40% of cases) and the esophago-gastric junction (approximately 16%). The neuroendocrine component was confirmed to be grade 3 in most cases and predominated in both the primary tumor and distant metastases. Moreover, the non-neuroendocrine component was present in histological evaluation in most cases of adenocarcinoma [203].

The study by Watanabe et al. showed the presence of MANECs of the colon and rectum in 3.2% of patients. The presence of this tumor had a worse prognosis for DFS and overall survival (OS) after curative resection compared with adenocarcinoma [204]. Studies also indicate that the presence of even less than 30% neuroendocrine component in colorectal tumors of a mixed glandular–neuroendocrine subtype can have a negative impact on the clinical course and patient outcomes, which makes the occasional finding of isolated NCs in CRC especially important [205]. In another study on the relatively large study population, adenocarcinoma with mixed subtypes was indicated as a rare (1.4%) but very aggressive histological subtype in CRC. Its prognosis was significantly worse than that of mucinous adenocarcinoma, but comparable to signet-ring cell carcinoma (SRCC). Mucinous adenocarcinoma, SRCC, and adenocarcinoma with mixed subtypes showed significantly poor survival rates compared with classical adenocarcinoma [206]. Colorectal MANEC is often diagnosed at an advanced stage when it is unresectable, and chemotherapy plays a major role in its treatment [207].

An interesting concept regarding the development and growth of CRC is the recognition of aberrant mechanisms in tissue dynamics of intestinal crypt epithelium understood as a polymer of cells. In steady state, they are maintained by an autocatalytic mechanism of polymerization. It is suggested that in patients with FAP and sporadic CRC, mutation in the APC gene increases autocatalytic colon tissue polymerization. It is hypothesized that when the autocatalytic polymerization reaction in colorectal tissues is enhanced, it may lead to progressive tissue growth and cancer development [208].

### 7.3. The SRIF System and Sporadic Colorectal Cancer Histogenesis

The association of the SRIF system with the histogenesis of sporadic CRC is also considered in research. It is known that APC mutations lead to CRC development by increasing the population of SCs. As it was mentioned, SCs and NCs are present in the SC niche of the colonic crypts, where the former mature into the latter. Thus, a possible mechanism for CRC initiation may be based on dysregulation of colonic NC maturation caused by APC mutations [11]. Computational analyses showed that APC mutations lead to reduced maturation of aldehyde dehydrogenase positive (ALDH)(+) SCs into progenitor NCs (rather than progenitor NCs into mature NCs) and reduced feedback of mature NCs. Moreover, it has been reported that mature NCs (GLP-2R(+)/SST1(+)), through their signaling peptides, exert opposing effects on the maturation rate of NCs via feedback regulation of progenitor NCs. However, this does not fully explain the delay in maturation, as both progenitor and mature NC numbers are lowered in CRCs. According to authors, CRC development results from an imbalance between Wnt signaling and retinoic acid (RA) signaling (with ALDH being a key component of the RA signaling). Generally, compared to normal colonic crypts, the number of NCs with specific markers (including SST) was reported to decrease in adenomas and carcinomas [11]. The interaction between SCs and NCs was also investigated, assuming that SST1 maintains SCs in a quiescent state. The authors tried to determine the way in which NCs affected by impaired SST signaling participate in SCs overpopulation. The lack of SST and SST1 expression on ALDH(+) stem cells suggested that SST signaling controls the maturation rate of NCs, as SCs mature into the NC lineage, which contributes to silencing of SCs and inhibition of proliferation. However, expression of both SST and SST1 was demonstrated in normal colorectal tissue, while only SST1 was expressed in CRC. In contrast, in CRC cell lines, the proportion of ALDH(+) cells was inversely correlated with the proportion of SST1(+) cells, as well as the rate of proliferation and sphere formation. Furthermore, each CRC cell line had a unique ALDH(+)/SST1(+) ratio that correlated with its growth rate. This suggests the existence of feedback mechanisms between SCs and NCs that participate in SC regulation [10].

## 8. Epigenetic Alterations of the SRIF System Components in Colorectal Cancer

Epigenetic regulatory mechanisms of SSTs and SSTR genes have been described primarily in NETs but are also relevant in other GI tract cancers, e.g., PDAC [139] and sporadic CRC [18,209,210,211,212]. Findings on epigenetic mechanisms in CRC suggest that hypermethylation of SST and SSTR genes may be a critical regulator in cancer development [18,209,211]. On the other hand, when it comes to the epigenetic machinery, special attention has recently been paid to the enzymes involved in modification of the expression of the entire SRIF system, with the results of such studies aiming to improve therapeutic approaches (reviewed in [49]).

Mori et al. demonstrated methylation of the SST gene in 88% of cases of CRC, and its severity was significantly higher in cancers with low-level microsatellite stability (MSI-L) than those without MSI-L. Moreover, this methylation was accompanied by a reduction in SST mRNA expression [209]. Interesting results were obtained by Leiszter et al., who showed a gradual increase in SST gene promoter methylation starting from juvenile colonic epithelium (3.5% ± 1.9%), through colonic epithelium in healthy adults (approximately 10%) up to developed CRC (30.2% ± 11.6%). These results suggested that an absence of local SST production may result in increased and unconstrained cell proliferation in CRC [18]. Subsequent studies have confirmed that increased gene methylation of both SST and SSTR2 is associated with their reduced expression in CRC. Furthermore, the SST gene was indicated as one of five hub genes, after constructing protein–protein interactions network for hypermethylation/low expression genes [211].

The role of the methylated SRIF system genes as epigenetic biomarkers in CRC has begun to be investigated and confirmed. Liu et al. demonstrated that methylation of serum SST gene can be an independent prognostic marker (for cancer-specific death and recurrence) in patients before tumor surgery [210]. Significant methylation of three other genes, including *SST2*, was also demonstrated in CRC tissues compared to adjacent normal colorectal tissue. The receiver operator characteristic (ROC) curve (AUC) value of *SST2* was 0.74, while a combination of three genes (*CMTM3*, *SST2,* and *MDFI)* produced an AUC value of 0.9 with a sensitivity of 81% [212]. Thus, in general, biomarkers based on aberrant methylation of genes belonging to the SRIF system, mainly *SST* and *SST2*, can be used to accurately diagnose and treat CRC.

In a taxonomy based on gene expression analysis, four “consensus molecular subtypes” (CMS) of CRC were distinguished: CMS1 (MSI Immune, 14%), CMS2 (Canonical, 37%), CMS3 (Metabolic, 13%), and CMS4 (Mesenchymal, 23%). Samples with mixed characteristics were also observed (13%). The latter may have represented a transient phenotype or heterogeneity within the tumor [213]. None of the identified molecular subtypes of CRC were clearly related to the SRIF signaling pathway, but the search for optimal markers for early diagnosis, prognosis, and more effective therapies for CRC, are still a challenge for science.

## 9. Clinical Application of the SRIF System Components

### 9.1. Diagnostics

The diagnosis and treatment of neuroendocrine tumors (especially well-differentiated NETs) with synthetic somatostatin analogues (SAAs) takes advantage of the fact that SSTRs (especially SST2, SST3, and SST5) are highly expressed in a high percentage of tumors [58,180,200,207,214,215,216,217]. The best target for the visualization of NETs is SST2 [215]. Diagnostic methods targeting this characteristic fall under the broader term of nuclear medicine. Nowadays, this approach bases on somatostatin receptor scintigraphy (SRS) and positron emission tomography/computed tomography (PET/CT) with ^68^Ga-labeled SSAs, which largely replaced SRS [175]. Overall, improvements in SSR (SSR/PET/CT) have increased the detection rate of NETs and led to the inclusion of this diagnostic method in an international strategy for patient management, from initial staging, through recurrence, to palliative care for almost all NETs [218].

A range of SAAs was described in detail, including their pharmacokinetic properties and therapeutic indications [216]. Many radioelements are used to label SSAs, such as indium-111 and technetium-99 m, and more recently, gallium-68, fluorine-18, and copper-64. Moreover, iodinated SST analogues, e.g., (111)-Indium (In)-labelled DTPA-octreotide, and (111)-In-pentetreotide (octreoScan) in SRS, were used for in vivo visualization of SSTR-positive tumors [111,214,215,219,220,221,222,223,224]. Octreotide scintigraphy is particularly useful for determining the status of SST2 and SST5 in diagnosed NET, aiding metastatic disease therapy targeting [200]. A positive correlation of ^68^Ga-DOTATOC uptake on dedicated PET/CT scanners with SST2 gene expression was demonstrated in NEC. Furthermore, a stronger correlation was demonstrated by SST2 than SST5 [225]. Recent studies in GEP-NETs indicate that imaging of SSTRs can be improved by administering an initial dose of unlabeled OCT just prior to radiotracer injection, simplifying the patient examination protocol [226].

A new dual classification scheme was also developed for diagnostic purposes—SomatoSTatin Receptor Imaging/Fluorine-18 (^18^F) deoxyglucose (SSTRI/FDG) (the NETPET grade)—allowing the characterization of the most metabolically active tumor based on FDG avidity relative to SSTRI avidity. The SSTRIs uptake was higher in well-differentiated than poorly differentiated NETs, strongly correlating with SST2A expression. The NETPET classification has been shown to be a promising prognostic biomarker in NETs. It captures the complexity of imaging with two radiotracers in a single disease-describing parameter and is easy to apply to patient management [227]. Others have confirmed the correlation between SSTR, FDG-PET, and tumor cellular differentiation, emphasizing that combined imaging is particularly useful in patients with a Ki-67 proliferation index >10% [228].

In turn, when it comes the clinical utility of SRS in the diagnosis of colorectal tumors (sporadic CRC, colorectal NETs, and mixed CRC), there are relatively few literature reports. OctreoScan seems to be particularly useful for evaluation of disease dissemination (detection of primary foci within the rectum is difficult) [222]. In Poland, [^99m^Tc-EDDA/hydrazinonicotinyl (HYNIC)]Tyr(3)-octreotide or ^99m^Tc-tectrotid, ^99m^Tc-EDDA/HYNIC-TOC or ^99m^HYNIC-TOC, and [^99m^Tc-EDDA/HUNIC]Tyr(3)-octreotate (^99m^TC-HYNIC-TATE) were also applied in this method [229,230,231,232]. In some European centers, gallium-labeled SAAs, ^68^Ga-[^68^Ga]DOTA-D-Phe^1^-Tyr^3^-Octreotide ^68^Ga-DO-TATOC, have been used for the diagnosis of highly differentiated lesions. However, this method had significant downsides (limited availability, high cost) [199,233].

Some studies using (111)-In-labelled DTPA-octreotide scintigraphy failed to demonstrate the receptor status of liver metastases in 10 CRC patients. This examination was not useful in planning therapeutic regimens, but the authors believe it may be helpful in histological differentiation of metastases of this tumor [214]. In contrast, in another study involving a case of a rectal carcinoid tumor, SRS was very useful in identifying the presence of lymph node metastases that were not captured by CT scanning [234]. The role of the fluorescent SSTR-specific conjugate 3207-86 in potential detection of CRC was also investigated. The study was conducted on human HT-29 cells induced in nude mice (mouse xenograft model of CRC). Increased detection of colorectal tumors by fluorescence imaging has been demonstrated, with a 5–8-fold increase in contrast between malignant and normal tissues [235]. Finally, according to Liepe et al., ^99m^Tc-TOC is a useful radiotracer in SSTR-expressing tumor lesion imaging, exhibiting marginally higher sensitivity, higher quality of imaging, and decreased patient radiation exposure compared to (111)-In-octreotide. However, false positive findings most probably resulted from the presence of colonic adenoma [232].

### 9.2. The SRIF System Components in Therapy of Cancers

Clinical application of somatostatin has been limited by its very short half-life (<3 min), necessitating continuous intravenous infusion. Hence, a number of SAAs with a longer half-life (1.5–2 h) are applied in radiolabeled somatostatin analogue/peptide receptor radionuclide therapy (PRRT) [23,25,26,58,217,236,237,238]. Moreover, while the native SST binds to all SSTRs, OCT binds with high affinity to SST2 and SST5 [239].

The most frequently used SSAs in PRRT clinical practice are OCT and octreotate (TATE). The latter has a higher affinity and selectivity towards SST2. Conjugation of both these peptides was performed to the 1,4,7,10-tetraazacyclododecane-1,4,7,10-tetraacetic acid (DOTA) bifunctional chelator, for either ^177^Lu or ^90^Y chelation [23,224,239,240].

Treatment with SSAs has become the gold standard therapy for primarily highly differentiated NENs of the GI tract, and other tumors that express SSTRs [45,107,214,217,241,242]. However, potential targets for molecular imaging and treatment in SSTR-negative NET are also reviewed [242]. Optimal therapy in NETs usually depends on the primary location and classification, morphologic features of differentiation, and tumor proliferation rates. Patients with highly differentiated (G1) hormonally active and hormonally inactive NENs, and/or slow rates of disease progression, and patients with high SSTR expression and low Ki-67 index are characterized with more favorable outcomes [107,123,187,243].

Therapy with SSAs takes advantage of both the multidirectional effects of SAAs on the GI tract (e.g., reducing secretion of hormones and biologically active substances) and their antiproliferative effects to reduce tumor mass, delay disease progression, and prolong life [26]. Long-acting release (LAR) OCT has been considered a breakthrough in the treatment of all NENs for about two decades [244,245]. It is a suitable first-line drug or maintenance therapy and can be used in combination therapy in advanced forms of NENs, and to delay disease progression. The demonstration of the antiproliferative effect of OCT LAR has led to its approval for the treatment of patients with NENs of unknown primary location (reviewed in [26]). Another long-acting SSA is lanreotide Autogel, whose use in the CLARINET trial significantly prolonged progression-free survival (PFS) in patients with metastatic pancreatic/intestinal NETs. The study proved that the antiproliferative effect was independent of liver metastasis [246]. In turn, results of the phase-three NETTER-1 trial determined high efficiency (longer PFS and a significantly higher response rate) and safety of ^177^Lu-DOTATE compared to high-dose OCT LAR in patients with advanced, progressive, SSTR-positive midgut (defined as jejunoileum and the proximal colon) NETs [236,247].

SAAs labeled with therapeutic radionuclides provide an overall response rate of approximately 30%. Hence, this treatment method continues to be improved by modifications of SSAs to increase blood circulation half-life and targeted accumulation in the tumor. One such modification was the use of DOTA-octreotate with Evans blue analogue. Using cell lines with different levels of SST2 expression, e.g., HCT116 (human colon cancer cells), HCT116/SSTR2, AR42J (rat amphicrinic pancreatic cells), and mice bearing SST2 xenografts, improved tumor response and survival rates and long-term efficacy was demonstrated compared to DOTA-octreotate itself [248]. Recently, action of nonpeptide 3,4-dihydroquinazoline-4-carboxamides as SST2 agonists was also described. This class of molecules exhibits high human SST2 potency and selectivity towards SSTRs. Hence, it can be used in carcinoids and NETs [249]. New therapeutic strategies consider the use of SST antagonist analogues, e.g., STT2-ANT (BASS), LM3, JR10, and JR11 (Satoreotide) [216,250]. Antagonist derivatives have been shown to be more effective compared with agonists, which may broaden the group of applications of these therapeutics (reviewed in [216]). The most current research findings and treatment guidelines for GEP-NETs are the subject of numerous publications [180,187,217].

#### Somatostatin Analogues in Neuroendocrine and Sporadic Colorectal Cancer Therapy

While NENs of colon and rectum (C-NENs and R-NENs) are generally rare, the frequency of their diagnosis is significantly increasing [198,200]. According to WHO classifications (2010, 2019), both these NENs are classified as Well-Differentiated Neuroendocrine Tumors (WD-NETs) that contain NET G1 and NET G2, and Poorly Differentiated Carcinomas (PD-NECs) comprising only G3 neoplasms (carcinomas) [123,186,198]. Colonic NETs occur in similar percentage of cases in the United States, Europe, and Asia (~7.5% of all NETs). Rectal lesions account for 18% of all NETs in the United States and 27% of all GI tract NETs, with slightly lower percentages reported in Europe (5–14% of the total GI tract NETs). Rectal NETs are more prevalent in Japan, making up 60–89% of all NETs of the GI tract [200].

High efficacy and morphologic responses with minimal toxicity and longer survival after PRRT were demonstrated in patients with metastatic NEN of the rectum, despite the unfavorable prognostic features of this cohort [251]. Interestingly, the use of OCT LAR and lanreotide Autogel in the above-label doses, in a group of 105 patients with various GEP-NETs, significantly prolonged the PFS of these patients. Additionally, patients with primary small intestine/colorectal cancers, Ki-67 index <5%, and no or limited extrahepatic metastases, benefited better from this therapy [217].

So far, it has been thought that SAAs may be difficult or even impossible to treat in sporadic CRC. Some authors have observed a loss of SST2 expression in more advanced stages of some CRC cases, or even in tumors of patients with higher CEA level, as compared with control or with patients with lower CEA levels, which could be associated with a loss of cellular regulation and subsequent excessive proliferation [117,138,166]. However, not all authors confirm the absence of SST2 expression in advanced morphological changes in CRC [115]. The presence of SST2 in CRC adenocarcinoma is also shown by own data (Figure 2). Demonstration of this SSTR subtype in CRC mostly bears significance due to OCT exhibiting high affinity for SST2, and lower for SST3 and SST5 [252]. Some published results also report antiproliferative and pro-apoptotic effects of OCT in colorectal carcinoma [171,174,214,217]. Recently, these effects of OCT, as well as better efficacy of cetuximab (CTX) combined with OCT, have been confirmed in the treatment of CRC. Furthermore, CTX–OCT conjugate was loaded onto Ca–alginate beads (CTX–OCT–Alg) and compared with single drug treatment. CTX was coated with alginate to enable its delivery into the GI tract [253]. According to other authors, an indirect antiproliferative effect of SSAs is also possible in tumors lacking SSTR expression. Hence, these peptides are most likely able to downregulate other growth promoting factors of the tumor (e.g., gastrin and insulin-like growth factor 1 (IGF-1)) [241].

Furthermore, different therapeutic options based on SSAs were tried in in vitro CRC models. Assuming that resistance of advanced forms of CRC to chemotherapy is often associated with the presence of *TP53* mutations in the tumor, the AN-238 analogue, consisting of 2-pyrrolinodoxorubicin (AN-201) linked to octapeptide SRIF carrier RC-12, was tested in CRC cell lines with wt p53 (HCT-116, LoVo cells), and p53 mutation (HCT-15, HT-29 cells). This therapeutic inhibited the growth of experimental colorectal cancers that express SSTRs, regardless of their p53 status [254]. Another experimental CRC model (HT-29, HCT-15, and HCT-116 cells) demonstrated the effect of a targeted cytotoxic SST analogue, AN-162, consisting of doxorubicin (DOX) conjugated to the SST carrier RC-121. In contrast to DOX alone, AN-162 blocked HCT-116 cells in the S/G2 phase and increased the number of apoptotic cells. In vivo, AN-162 reduced CRC xenograft volume more effectively than its non-conjugated components [170]. Promising results of in vitro studies conducted on HCT116 cells were also obtained during the evaluation the ^64^Cu-CB-TE2A-Y3-TATE SST2 agonist in p53- and SST2-positive tumors [255]. Recently, the mechanisms of action of newer forms of therapy, based on SST radio antagonists (^177^Lu-labeled SST), have been investigated in similar in vitro models (HT-29 cells). ^177^Lu-DOTA-Peptide 2 exhibited high stability in vitro and good SSTR affinity. The acceptable uptake of this SSA by tumors and the high tumor-to-blood ratio of ^177^Lu-DOTA-Peptide 2 may allow the introduction of this radiopeptide as a therapeutic agent for colorectal adenocarcinoma in humans [256].

In summary, SSAs are the subject of intense research. They are a good therapeutic option in NENs (including colorectal NENs) compared to other more toxic anti-cancer therapies. Novel peptide and non-peptide SAAs with a binding profile similar to native SST are sought [257]. It is also a challenge to describe the crystal structure of the five SSTRs to facilitate the discovery of new, safe, and effective agonists and antagonists of SRIF system components [12,180,255]. Furthermore, non-peptide agonists of SSTRs may provide the basis for the development of novel oral antidiarrheal therapies [257].

The challenge in NENs therapy is also to determine the optimal forms of combined therapies, integrating SAAs with chemotherapy or molecular therapies, and expanding the indications for SAAs in other cancers. Which of these SRIF property-based therapy systems will be able to be used in sporadic CRC will most likely become clear in the near future.

## 10. Conclusions and Future Challenges

There are many excellent and accessible literature positions on the role of the SRIF system in GI tract in physiology and disease. However, this manuscript aims to present the lesser known role of the SRIF system in histogenesis, diagnosis, prognosis, and therapeutic approaches in neuroendocrine and sporadic CRC.

In the GI tract, most studies have investigated the role of SRIF system in the physiology and pathology of various NENs (including colon and rectum NENs). The contribution of the SRIF system to the development, diagnosis, and prognosis of sporadic CRC is still being studied and discovered. It is suggested that neuroendocrine carcinoma (NEC) may arise from preceding adenocarcinomas. In CRC histogenesis, the role of the SRIF system (mainly SST and SST1) is linked to APC gene mutations and could involve dysregulation of feedback mechanisms between cancer stem cells and neuroendocrine cells present in the niches of intestinal crypts. Epigenetic changes (mainly methylations) of SRIF system components (mainly SST and SST2) may be helpful in the diagnosis and prognosis of CRC, as well as serve in better treatment qualification. It was demonstrated a progressive increase in SST gene promoter methylation starting from juvenile colonic epithelium and ending on tumor-transformed epithelium in CRC. SST methylation is described in nearly 90% of CRC patients. This is accompanied by a loss of antiproliferative function of the SRIF system and tumor growth. On the other hand, the methylated forms of SST and SST2 were determined as sensitive prognostic markers.

Determination of serum SST levels as a potential marker of endocrine tumor differentiation (including CRC) does not provide conclusive results. A recent retrospective audit of fasting gut hormone profile indicates that elevated SST levels (a concentration >150 pmol/L was considered abnormal) were observed in 5 out of 231 patients with suspected NENs, of which only two patients were diagnosed with pancreas NET. Three patients had false positive elevated plasma concentrations of fasting SST. The reasons for these abnormal results are discussed [258].

In the diagnosis of colorectal tumors (including sporadic CRC), the use of somatostatin receptor scintigraphy (SRS) can be useful to assess disease dissemination, histologically differentiate tumor metastases, and identify nodal metastases not visible by other methods. However, its widespread use still applies almost exclusively to tumors with an endocrine component.

It remains to be seen whether the treatment of sporadic CRC based on SRIF system components (SST agonists or antagonists) will be clinically effective. So far, various therapeutic options are being evaluated using different colon cancer cells in vitro. Potential therapeutics with promising anti-tumor effects include targeted cytotoxic SSAs in combination with chemotherapeutics, SST2 agonists, and SST antagonists.

There are still many questions in the research on the biological role of the SRIF system. In the context of the contribution of the SRIF system in CRC pathogenesis, progression, and therapy, several questions can be highlighted: (1) Which factors regulating somatostatin secretion in the GI tract (intraluminal, neural, and endocrine) predominate in healthy colon, and which are dysregulated in CRC? (2) What is the potential relationship between the number of colorectal EECs, hormonal spectrum (including SRIF system), and dietary nutrient exposure in CRC pathogenesis? (3) What are the molecular mechanisms regulating the differentiation of CRC subtypes involving the SRIF system? (4) What are the precise mechanisms of immune system regulation in CRC? (5) What exactly is the significance of SST secreted by enteric nerve system elements in the course of CRC? (6) Which peptide derivatives are effective in the treatment of sporadic CRC with lower SSTR expression?

## Figures and Tables

**Figure 1 biomedicines-09-01743-f001:**
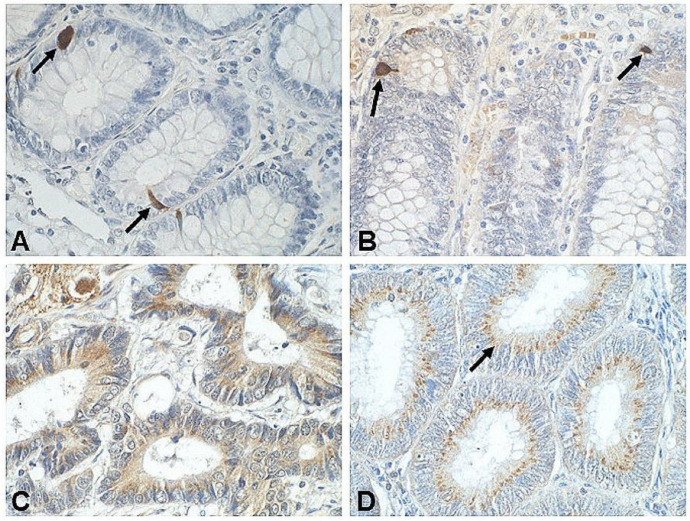
(**A**) Representative image of the immunohistochemical (IHC) detection of human 5-hydroxytryptamine (5-HT, serotonin) in individual enterochromaffin cells (ECs) in normal colon mucosa (arrows). (**B**) Somatostatin-immunoreactive endocrine cells located between the epithelial cells lining the normal colon mucosa in the same patient (arrows). (**C**,**D**) Somatostatin expression in colorectal adenocarcinoma tissue samples. Worth noting is the homogeneous IHC reaction pattern in the whole cytoplasm of the cells (**C**) and a granular IHC reaction in the apical part of the cell (arrow) (**D**). New polymer-based immunohistochemistry with DAB staining. Hematoxylin counterstained. Original magnification ×400 (own unpublished observations).

**Figure 2 biomedicines-09-01743-f002:**
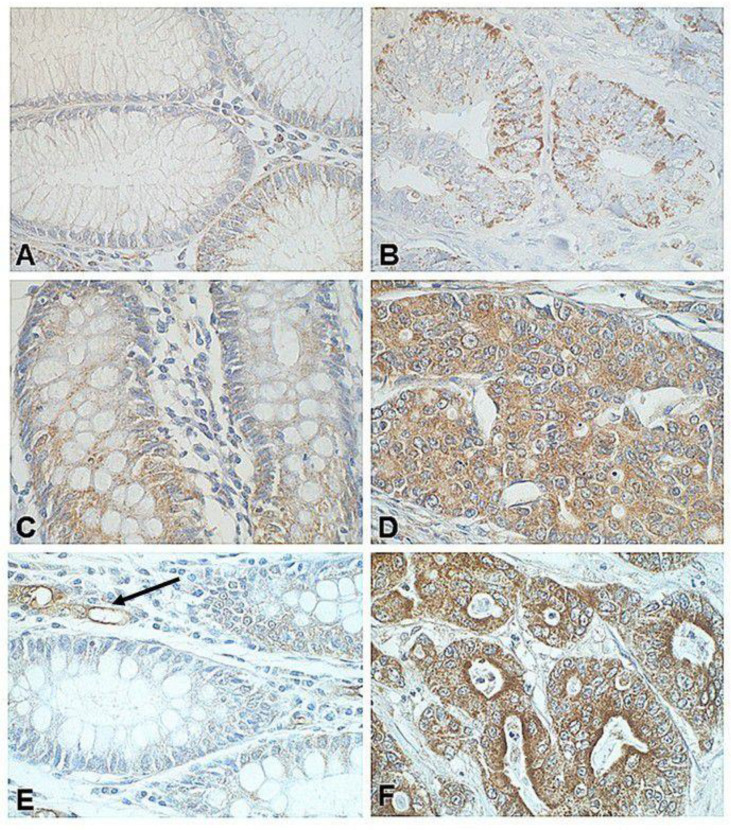
Representative images of the immunohistochemical detection of STT2, SST3 and SST5 as shown by brown staining of the crypt cells in normal colon mucosa and CRC cells forming glandular structures. (**A**,**C**,**E**) SST2-, SST3-, SST5-immunoreactive cells in normal colon mucosa, respectively; (**B**) Membranous pattern of IHC reaction for SST2 in CRC tissue sample. (**D**,**F**) Cytoplasmic SST3- and SST5 expression in CRC tissue samples, respectively. Worth noting is the positive IHC reaction for SST5 also on endothelial cells and/or vascular smooth muscle cells (**E**) (arrow). New polymer-based immunohistochemistry with DAB staining. Hematoxylin counterstained. Original magnification ×400 (own unpublished data).

**Figure 3 biomedicines-09-01743-f003:**
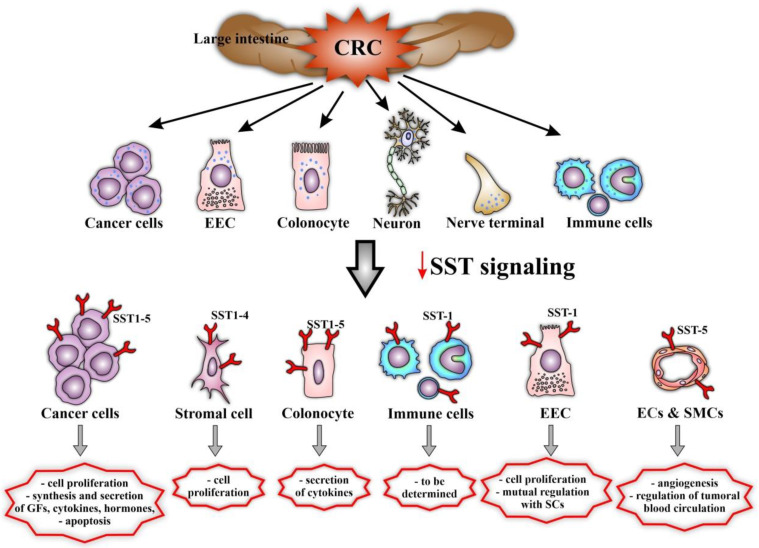
Cellular sources of somatostatin (SST) with its local activity via somatostatin receptors 1–5 (SST1–5) present on different cells in colorectal cancer (CRC). The reduced secretion of SST and the highly heterogenous expression of SST1–5 on tumor cells results in impaired primary biological effects of the SRIF system in patients with colorectal cancer (for details see text). ↓—reduced expression/activity; ECs—endothelial cells; EEC—enteroendocrine cell; GFs—growth factors; SCs—stem cells; SMCs—smooth muscle cells.

**Table 1 biomedicines-09-01743-t001:** Molecular characteristics of the human SRIF system components and the major biological effects in the gastrointestinal (GI) tract.

SRIF System Member	Gene Location and Size (kb)	No. of Transcripts	Protein	Mechanisms of Action/Biological Effects in GI Tract	Ref.
SST	*SS1* gene, 3q27.3; size: 1483 bases	pre-mRNA contains an intron flanked by two exons; 1 transcript (splice variant) and 262 orthologues	preprosomatostatin (116 AA) and prosomatostatin (92 AA), which in turn is C-terminally processed to generate the cyclic peptides SST-14 AA and SST-28 AA (N-terminal extension); m.m. 12736 Da; q.s.: nd	(+)SST1–5 (GPCRs family); (−)several hormones secretion; (−)exocrine secretion; (−)gallbladder contraction; (−)small intestinal segmentation; (−)gastric emptying;regulation of gastric acid, digestive enzymes, bile, colonic fluid, and motility; local immunomodulatory actions	[12,28,32,37,41,60,61,62,63,65,66,75,88,90,91]
CST	*CORT* gene; 1p36.3–1p36.2; size: 1968 bases		cleavage of preproCST (112 AA) gives rise to multiple mature products, CST-14, CST-17 and CST-29; m.m. 11532 Da, q.s.: nd	CST may activate SST1–5 and GPCRs other than SST1–5; share several biologic properties with SST	[12]
SST1	14q13; Size: 5164 bases	an intronless gene	391 AA; m.m. 42686 Da; q.s.: interacts with SKB1	binds both form of SST, with slightly higher affinity for SST-14; (−)proliferation; (−)secretion; (−)intestinal Cl^−^ secretion	[12,27,62,106,107]
SST2	17q25.1; size: 11,624 bases	consists of two exons	369 AA; spliced into SST2A and 2B; human tissues exclusively contain the unspliced SST2A variant; m.m. 41333Da; q.s.: homo- and heterodimer with SST3 and SST5	binds both forms of SST, with slightly higher affinity for SST-14; (−)proliferation; (−)secretion; (+)apoptosis; (−)exocrine secretion: gastric acid and intestinal Cl^−^ secretion	[12,27,62,106,107]
SST3	22q13.1; size: 16,215 bases	spreads over eight exons	418 AA; m.m. 45847 Da; q.s.: homodimer and heterodimer with SST2	binds both forms of SST, with slightly higher affinity for SST-14; (−)proliferation; (−)secretion; (+)apoptosis; gastric and intestinal relaxation	[12,27,62,102,103,106,107]
SST4	20p11.2; size: 3926 bases		388 AA; m.m. 42003 Da; q.s.: nd	binds both forms of SST, with slightly higher affinity for SST-14; (−)proliferation; (−)secretion	[12,27,62,104,106,107]
SST5	16p13.3; size: 8708 bases		364 AA; m.m. 39202 Da; q.s.: heterodimer with SST2	binds both forms of SST, with a 10-fold higher affinity for SST-28; (−)proliferation; (−)secretion, e.g., amylase secretion; (−)colonic contraction	[12,27,62,104,106,107]

Legend: (+)/(−)—activation/inhibition; AA—amino acid; CST—cortistatin; GPCR—G protein-coupled receptors; m.m.—molecular mass; nd—no data available; q.s.—quaternary structure; ref.—number of references; SKB1—protein arginine N-methyltransferase skb1; SST/SRIF—somatostatin/somatotropin release–inhibiting factor; SSTRs/SST1–5—somatostatin receptors 1–5.

**Table 2 biomedicines-09-01743-t002:** Tissue expression of somatostatin (SST) and its receptors (SST1–5) in different subtypes of colorectal cancer (CRC).

SRIF System Member	Cellular Localization	The Main Results	Material and Methods	Ref.
SST	tumor cells	(+)in 18/23 tumors	*n* = 32 NETs of rectum (*n* = 27 as typical carcinoids); IHC	[163]
	tumor cells	(+)in 3% tumors	*n* = 3 carcinoids of the distal sigmoid colon, and *n* = 81 of the rectum; IHC	[164]
	tumor endocrine cells	(+)in 10% of chromogranin A-positive CRC	*n* = 350 CRC; IHC	[160]
	carcinoma cells	(+)in 82% CRC vs. 90% tumor-neighboring mucosa	*n* = 100 CRC (39 colonic and 61 rectal), and surrounding mucosa; IHC	[8]
	tumor cells; normal colonocytes	(+)in 84.6% CRC vs. 88.5% tumor-neighboring mucosa	*n* = 53 advanced CRC, and tumor-neighboring mucosa; IHC	[9]
	tumor EECs in glandular structures	(+)in 4/57 CRC; in 2 cases co-expressed with serotonin and glucagon	*n* = 57 CRC; IHC	[55]
	carcinoma cells	↑in well-differentiated vs. poorly differentiated tumors; ♣(↓expression)	*n* = 35 CRC; *n* = 25 LM; IHC	[165]
	D cells	low expression	*n* = 90 mirror biopsies of CRC; iEM	[161]
	carcinoma cells	(−)expression	*n* = 1 mucinous CRC with signet-ring cells (stage D); IHC; iEM	[57]
	ENS structures	frequency of (+)neurons/fibers in the intact vs. CRC areas*^NS^*	*n* = 15 CRC; IHC; IF	[162]
	D cells; tumor cells	↓mRNA in CRC vs. C (adults); ↑ratio of (+)cells in C (children) vs. CRC	*n* = 34 CRC, *n* = 6 C (children), *n* = 41 C (adults) (TMA); *n* = 13 CRC, *n* = 14 C (children), *n* = 20 (adults) (IHC); *n* = 12 CRC, *n* = 12 C (children), *n* = 12 C (adults) (RT-PCR)	[18]
		(+)in all the C; (−)in matching CRC samples	5 samples of C, and matched CRC; RT-PCR	[10]
SST1		heterogeneously expressed in both C and CRC; (+) in liver M	*n* = 47 CRC and C; RT-PCR	[166]
	tumor cells; normal colonocytes	expressed infrequently	*n* = 32 pairs of CRC and C; RT-PCR; ISH	[167]
		(+)mRNA frequently expressed in C and CRC	CRC and C; RT-PCR	[115]
	immune cells of the lp close to the tumor; stromal cells; epithelial cells	(+)mRNA expressed in C and CRC	CRC and C; ISH; image analysis system	[116]
	tumor cells; normal colonocytes	the predominant subtype in CRC and C; ♦↑(Dukes stage); ♦↑(lymph node M)	*n* = 127 CRC and *n* = 40 C; IHC	[118]
		(+)in all the C; (+)in 3/5 CRC	5 samples of C and matched CRC tumor tissue; RT-PCR	[10]
SST2		(+)mRNA (50% in Dukes stage B, and 20% in Dukes stage C), (−)in stage D CRC	*n* = 47 CRC and controls, RT-PCR	[166]
	tumor cells; normal colon crypt cells	(+)in nearly 90% CRC and C	*n* = 32 pairs of CRC and C; RT-PCR; ISH	[167]
		(+)mRNA frequently expressed in C and CRC; no loss in advanced CRC stages	CRC and C; RT-PCR	[115]
	tumor cells; immune cells; epithelial cells; stromal cells	low mRNA expression in C and CRC	CRC and C; ISH; image analysis system	[116]
		(+)in 100% tumors; CRC vs. C*^NS^*; loss of SST2 mRNA in patient’s tumor with ↑CEA level vs. patients with low CEA; normal/tumor mRNA ratio inversely related to CEA levels	*n* = 26 CRC and corresponding C; qRT-PCR	[138]
		(+)in 100% tumors; CRC vs. C*^NS^;* loss of SST2 mRNA in tumor in patients with ↑CEA level vs. C	*n* = 100 CRC and corresponding C; RT-PCR; ISH	[117]
		↑mRNA; # (↑CRD and shorter DFS)	qRT-PCR	[54]
	tumor cells; normal colon crypt cells	the second subtype in C and CRC; ↑in moderately to well vs. poorly differentiated CRC; ↓in the ulcerative type of CRC	*n* = 127 CRC and *n* = 40 C; IHC	[118]
	tumor cells	↑in lower-grade and rectum located tumors; ♦, #, (+) (longer survival rate); ♦,↓liver M	*n* = 81 CRC; IHC	[119]
	tumor cells	(+)in 100% tumors	*n* = 3 rectal NETs; IHC	[120]
		(−)SST2A	*n* = 19 colorectal MANECs and *n* = 8 CRC poorly differentiated NETs; IHC	[52]
	tumor cells	(+)expression	*n* = 1 small cell anal NEC; IHC	[168]
SST3		rarely or not expressed	*n* = 47 CRC and C; RT-PCR	[166]
	tumor cells; normal colonocytes	expressed infrequently	*n* = 32 pairs of CRC and C; RT-PCR; ISH	[167]
	tumor cells; transformed epithelial cells; stromal cells	low mRNA expression in C and CRC	CRC and C; ISH; image analysis system	[116]
	tumor cells; normal colon crypt cells	♦,↓(Dukes stage)	*n* = 127 CRC and *n* = 40 C; IHC	[118]
SST4		heterogeneously expressed in C and CRC; (+) in liver M	*n* = 47 CRC and C; RT-PCR	[166]
	tumor cells; normal colonocytes	expressed infrequently	*n* = 32 pairs of CRC and C; RT-PCR; ISH	[167]
	stromal cells; immune cells in CT close to the tumor; epithelial cells	low mRNA expression in both CRC and C	CRC and C; ISH; image analysis system	[116]
	tumor cells; normal colonocytes	↑in moderately to well vs. poorly differentiated CRC; ↑frequency in tumor cells (18.9%) vs. C (2.5%)	*n* = 127 CRC and *n* = 40 C; IHC	[118]
SST5		heterogeneously expressed in C and CRC	*n* = 47 CRC and C; RT-PCR	[166]
	tumor cells; normal colonocytes	(+)in 46% CRC and 45% C; (+)in 75% (Dukes A and B stage) vs. 31% (Dukes C and D); ↓in M (11%) vs. all tumor samples (56%)	*n* = 32 pairs of CRC and C; RT-PCR; ISH	[167]
		(+)mRNA very frequently expressed in C and CRC; no loss in advanced CRC stages; ↑frequency in the left CRC vs. C	CRC and C; RT-PCR	[115]
	tumor cells; epithelial cells; stromal cells	(+)the mRNA was the predominant subtype expressed in C and CRC; ↑in CRC vs. C	CRC and C; ISH; image analysis system	[116]
	tumor cells; normal colonocytes	the second subtype in C and CRC; ↑in moderately to well vs. poorly differentiated CRC; ♦,↓(Dukes stage)	*n* = 127 CRC and *n* = 40 C; IHC	[118]
	tumor cells	(+)(longer survival rate); ↓, #, liver M	*n* = 81 CRC; IHC	[119]
	tumor cells	(+)in 66.6% tumors	*n* = 3 rectal NETs; IHC	[120]

Legend: (+)/(−)—positive/negative expression/correlation; ↑/↓—significant increased/decreased expression; ♣—significant association between SST/SSTRs expression and degree of cancer differentiation; ♦—association between SST/SSTRs expression and more advanced clinical stage of cancer (Dukes stage, TNM, tumor size, venous infiltration, microsatellite nodules, metastases, etc.); #—significant correlation with poor prognosis (CRD, DFS); C—control, normal colon mucosa; CEA—carcinoembryonic antigen; CRC—colorectal carcinoma; CRD—cancer-related death; CSS—cancer-specific survival; DFS—disease free survival; EECs—enteroendocrine cells; ENS—enteric nervous system; iEM—immunoelectron microscopy with immunogold staining; IF—immunofluorescent microscope; IHC—immunohistochemistry; ISH—in situ hybridization; lp—lamina propria; MANEC—mixed adenoneuroendocrine carcinoma; M—metastasis; *n*—number of cases; NE(C,T)—neuroendocrine (cancer, tumor); NS—non significant; nt—not tested; PD—paraformaldehyde; qRT-PCR—quantitative real-time PCR; ref.—number of references; RIA—radioimmunoassay; SST/SRIF—somatostatin/somatotropin release–inhibiting factor; SSTRs/SST1–5—somatostatin receptors 1-5; TMA—tissue microarray; TNM—tumor, node, metastasis.

## Data Availability

Not applicable.

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
