# Peer review of "Somatostatin and Its Receptor System in Colorectal Cancer"

_biomedicines, 2021, doi:10.3390/biomedicines9111743_

Round 1
Reviewer 1 Report
This is a review on the role of somatostatin system in colorectal cancer. The problem is not often reviewed and the present article is extensive and definitive. The references are adequate and the manuscript is well organized.
Author Response
Dear Reviewer,
I wish to thank you very much for a review, and time spent on reviewing the manuscript. Thank you and I really appreciate such a favourable review of my work.
Reviewer 2 Report
The manuscript deals with plenty of literature in order to give an extensive and detailed review about the physiology, pathology and targeting – for both diagnostic and therapeutic purposes – of the Somatostatin and its receptors in colorectal cancer. Each aspect is carefully dissected with examined. However, few minor corrections have to be made before its acceptance. In detail:
- The acronym NENs/NETs is abbreviated repeatedly in different parts of the manuscript (line 69, line 256, line 488). Please correct and check carefully if other abbreviations are also repeated.
- Table 1 is outside of the margins and therefore readable only in part, please fix it
- The sentence in lines 572-576 is pretty obscure, it should be explained more in detail
- Line 636: it’s copper not cooper
Author Response
Dear Reviewer,
I wish to thank you very much for a favourable review, all critical remarks and time spent on reviewing the manuscript.
I have tried to review and correct all acronyms of NENs/NETs, and have removed repetitions of the entire name in some places (e.g. line 269). Changes to the NETs/NENs terminology are presented in a separate section of the paper (chapter 7). Terms used by the authors have been left in the works cited.
Posting Table 1 and Table 2 for technical reasons was difficult for me, but I was able to correct this fault as well.
I corrected a little bit misunderstood sentence in lines 572-576, and an error in line 636.
The publication was corrected also by a qualified, native speaker, familiar with the manuscript topics.
All changes (and all additions) in the text were marked red.
Reviewer 3 Report
The author wrote an extensive review about the role of Somatostatin in CRC. It explains the SRIF system in general, in different tumor types, the large intestine, IBD, CRC, CRC histogenesis, epigenetic alterations of SRIF, and the clinical applications. The review contains a fair amount of references including current references.
I have some points of attention that might help to further improve the manuscript:
- The title contains a research question. However, the answer to this question is not present in the abstract nor in the conclusion paragraph.
- Figure 2 seems a bit off in this review manuscript since it is not referring to existing work but shows unpublished own data instead. I think it is an editorial decision whether it can remain in the manuscript or should be excluded.
- Table 2 addresses the expression of somatostatin and receptors in subtypes of CRC. Is there any way to include the relationship with the molecular subtypes of CRC as described by Guinney et al., (Nature, 2015)?
- Could you please elaborate in the manuscript text on the sentence in figure 3 “- to be determined” for the effect of SST on immune cells”? Also more general, there is no reference toward figure 3 in the main text.
In general, the manuscript is thorough, structured, and in general, well written. However, the readability could possibly be improved.
- I would like to suggest reordering the manuscript. Although the current order makes sense from a biological perspective, most of the readers would probably start at section 6 since this is where the title is referring to and the prior sections could be useful as background information. Section 10 summarizes the most informative sections in the sentence describing the aim of the study ” histogenesis, diagnosis, prognosis, and therapeutic approaches”
- the readability could possibly be improved by adding signal words. For example on Page 5: ” …three groups of factors: intraluminal, neural, and endocrine. The first group … Adding ‘second’ and third / final starting the paragraphs about neural factors and endocrine factors could guide the reader.
- Furthermore, especially writing large paragraphs, end with a summarizing sentence like “In conclusion,…”, “In summary,…” etc.
Author Response
Reviewer 3:
I wish to thank you very much for a favourable review, all critical remarks and time spent on reviewing the manuscript.
- Thank you for your comment regarding the title of the paper. Since it could indeed be misunderstood by readers, it has been shortened to the original version.
- Ad Figure 2 - The purpose of Figure 2, like Figure 1 in this review manuscript, is solely to illustrate the cellular localization of the components of the SRIF system performed for the purposes of this work. Since the results of such studies mostly corroborate the studies of other authors in this area, I have dared to include these microphotographs.
- The Guinney et al. (2015) paper suggested by the reviewer was cited, although it is indirectly related to the topic of the paper.
- For the description to Figure 3, I incorrectly quoted Figure 1 in the text instead of Figure 3, for which I apologise. This has been changed. I completed data on the potential role of SRIF system components expression on immune cells in inflammatory bowel diseases and CRC.
The order of the chapters of the paper has not been changed because the structure of the article adopted by me seems to be the one most often adopted in review papers. All background information is important to proceed to such studies in colorectal cancer. In citations, an attempt has been made to retain the original wording of some CRC subtypes studied by the authors, although it is known that the taxonomy of cancer is changing dynamically and probably today the names of CRC are more appropriate to the observed pathological changes and also enriched by the description of molecular features.
Larger subsections on CRC and SRIF system were concluded with summaries, e.g. 6.1., 6.2. (including Figure 3) and 9.0. However, heterogeneity of data in some subsections did not always allow for such a summary.
As suggested by the reviewer, I emphasized signal words in the indicated places. The description of each factor is left in separate paragraphs.
As per suggestion, the manuscript was thoroughly revised, with all language errors corrected. The publication was corrected by a qualified, native speaker, familiar with the manuscript topics.
All changes (and all additions) in the text were marked red.